# The Effect of Sun Tan Lotion on Skin by Using Skin TEWL and Skin Water Content Measurements

**DOI:** 10.3390/s22093595

**Published:** 2022-05-09

**Authors:** Perry Xiao, Daqing Chen

**Affiliations:** School of Engineering, London South Bank University, London SE1 0AA, UK; chend@lsbu.ac.uk

**Keywords:** capacitive imaging, machine learning, skin water content, sun tan lotions, trans-epidermal water loss

## Abstract

Stratum corneum (SC) is the outermost skin layer. SC hydration is important for its cosmetic properties and barrier function. SC trans-epidermal water loss (TEWL) measurements and skin water content measurements are two key indexes used for SC characterisation. The instrument stability and accuracy are vitally important when measuring small changes. In this paper, we present our latest study on the effect of sun tan lotion on skin by using skin TEWL and skin water content measurements. We developed techniques to improve the measurement stability and to visualise small changes, as well as developed machine learning algorithms for processing the skin capacitive images. The overall results show that TEWL and skin water content measurements are capable of measuring the subtle changes of skin conditions due to the application of sun tan lotions. The results show that the TEWL values decreased after the sun tan lotion application. The sun tan lotion with SPF 20 had the lowest decrease, whilst the sun tan lotion with SPF 50+ had the highest decrease. The results also show that the skin water content increased after the sun tan lotion application, with SPF 20 having the highest increase, whilst SPF 50+ had the lowest increase.

## 1. Introduction

Stratum corneum (SC) hydration is important for its cosmetic properties and barrier function. SC trans-epidermal water loss (TEWL) measurements and skin water content measurements are two key indices used for SC characterisation [1,2]. Talor investigated occupational skin health by using TEWL measurements [3]. Mutai et al. studied the effects of skin moisturiser by tape-stripping and TEWL measurements on the heel of healthy, young adults [4]. Kis et al. studied effect of non-invasive dermal electroporation on skin barrier function and skin permeation in combination with different dermal formulations, using TEWL measurements for skin barrier function evaluation [5]. Uchegbulam et al. studied the effect of seasonal change on the biomechanical and physical properties of the human skin, finding an inverse relationship between TEWL and the average epidermal roughness (AER) [6]. Denzinger et al. conducted a quantitative study of TEWL on conventional and microclimate management capable mattresses and hospital beds [7].

Our previous studies have shown that capacitance-based fingerprint sensors, originally designed for fingerprint biometric measurements, can be adapted for skin hydration imaging, surface analysis, 3-D surface profile, and skin micro-relief measurements [8,9,10]. Pan et al. studied the occlusion effects in capacitive contact imaging for in vivo skin damage assessments [11]. Ou et al. conducted an in vivo skin capacitive imaging analysis by using the Grey Level Co-occurrence Matrix (GLCM) algorithm [12]. Xiao et al. studied membrane and pig skin solvent penetration by using skin capacitive imaging [13]. Zhang et al. used capacitive imaging for skin characterisations and solvent penetration measurements [14]. Bontozoglou et al. conducted skin micro-relief analysis with skin capacitive imaging [15]. Elsewhere, Navaraj et al. developed fingerprint-enhanced capacitive-piezoelectric flexible sensing skin to discriminate static and dynamic tactile stimuli for robotic arms [16]. Multispectral fingerprint biometric systems have also recently become quite popular as they provide high security and recognition [17,18].

In this paper, we present our latest study on the effect of sun tan lotion on skin by using skin TEWL measurements and skin water content measurements. The purpose of this study was to develop a new SC barrier characterisation method by using both TEWL and capacitive imaging measurements in order to measure subtle changes in the skin status. We first present the theoretical and technical background, then the results and discussions. The results show that precise measurements of skin TEWL and skin water content measurement can reveal otherwise undistinguishable changes in skin condition.

## 2. Materials and Methods

This section describes the skin measurement devices used, the machine learning algorithms developed, the volunteer information, and the measurement procedures. Two TEWL measurement devices were used in order to see the effect of instrument variations on measurement results.

### 2.1. Skin Measurement Devices

Figure 1 shows photos and schematic diagrams of the Epsilon permittivity imaging system and the AquaFlux TEWL instrument (Biox Systems Ltd., London, UK). The Epsilon is based on a Fujistu fingerprint sensor, which has 256 × 300 pixels with 50 mm spatial resolution and 8-bit grey-scale capacitance resolution per pixel [10,11,12,13,14,15]. As a contact technology, consistent contact is a key to the measurement repeatability and accuracy. Several approaches have been adapted to ensure the consistent contact, a spring loaded mechanism of the sensing area, measurement starting threshold, and starting delay. With the Epsilon, we can measure skin surface hydration dynamically during occlusive contact in order to generate time-dependent grey-scale occlusion curves. 

The AquaFlux uses the closed condenser-chamber measurement method [19,20,21]. Its cylindrical measurement chamber is open at the end placed onto the skin surface and closed at the other end by means of a condenser cooled below the freezing point of water. This design provides a controlled measurement environment, which enhances the repeatability and accuracy of the measurements. With the AquaFlux, we can accurately measure TEWL. AquaFlux and Epsilon were chosen for this study due to their high accuracy and high repeatability [22,23]. An alternative TEWL device, the VapoMeter (Delfin, Finland), is also used in the study for comparison reasons.

### 2.2. Machine Learning Algorithms

Several machine learning algorithms have also been developed for processing skin capacitive images. A new region of interest (ROI) searching algorithm based on template matching [24,25] has been developed, with which we can select a ROI from one image and search and locate exactly the same region in the consequent images. 

Six different searching methods were used in order to obtain the best matching results: squared differences (SqDiff), normalised squared differences (SqDiff_Normed), cross correlation (CCorr), normalised cross correlation (CCorr_Normed), correlation coefficient (CCoeff), and normalised correlation coefficient (CCoeff_Normed). By using ROI, we can improve the accuracy when analysing a sequence of images. 

If we use ***T***(***x***,***y***) to represent the ROI, where ***x*** and ***y*** are the horizontal and vertical positions in the image, respectively, and using ***I***(***x***,***y***) to represent the target image, then we can use the following equation to calculate the matching result image ***R***(***x***,***y***) for the squared differences (SqDiff) method: (1)$$R\left(x,y\right)=\sum_{x^{\prime },y^{\prime }}{\left(T\left(x^{\prime },y^{\prime }\right)-I\left(x+x^{\prime },y+y^{\prime }\right)\right)}^{2}$$

We can also use the following equation to calculate the matching result image ***R***(***x***,***y***) for normalised squared differences (SqDiff_Normed):(2)$$R\left(x,y\right)=\frac{\sum_{x^{\prime },y^{\prime }}{\left(T\left(x^{\prime },y^{\prime }\right)-I\left(x+x^{\prime },y+y^{\prime }\right)\right)}^{2}}{\sqrt{\sum_{x^{\prime },y^{\prime }}{\left(T\left(x^{\prime },y^{\prime }\right)\right)}^{2}*\sum_{x^{\prime },y^{\prime }}{\left(I\left(x+x^{\prime },y+y^{\prime }\right)\right)}^{2}}}$$

For SqDiff and SqDiff_Normed methods, the location (***x,y***) in ***R***(***x***,***y***) that has the minimum value is the position of the best match. 

Similarly, we can use the following equation to calculate the matching result image ***R***(***x***,***y***) for cross correlation (CCorr):(3)$$R\left(x,y\right)=\sum_{x^{\prime },y^{\prime }}{\left(T\left(x^{\prime },y^{\prime }\right)*I\left(x+x^{\prime },y+y^{\prime }\right)\right)}^{2}$$

For normalised cross correlation (CCorr_Normed):(4)$$R\left(x,y\right)=\frac{\sum_{x^{\prime },y^{\prime }}{\left(T\left(x^{\prime },y^{\prime }\right)*I\left(x+x^{\prime },y+y^{\prime }\right)\right)}^{2}}{\sqrt{\sum_{x^{\prime },y^{\prime }}{\left(T\left(x^{\prime },y^{\prime }\right)\right)}^{2}*\sum_{x^{\prime },y^{\prime }}{\left(I\left(x+x^{\prime },y+y^{\prime }\right)\right)}^{2}}}$$

For correlation coefficient (CCoeff):(5)$$R\left(x,y\right)=\sum_{x^{\prime },y^{\prime }}{\left(T^{\prime }\left(x^{\prime },y^{\prime }\right)*I^{\prime }\left(x+x^{\prime },y+y^{\prime }\right)\right)}^{2}$$
where
(6)$$T^{\prime }\left(x,y\right)=T\left(x^{\prime },y^{\prime }\right)-\frac{1}{w*h}\sum_{x^{\prime \prime },y^{\prime \prime }}T\left(x^{\prime \prime },y^{\prime \prime }\right)$$

Here, ***w*** and ***h*** are the width and height of the image ***T***(***x***,***y***), respectively, and:(7)$$I^{\prime }\left(x+x^{\prime },y+y^{\prime }\right)=I\left(x+x^{\prime },y+y^{\prime }\right)-\frac{1}{w*h}\sum_{x^{\prime \prime },y^{\prime \prime }}I\left(x+x^{\prime \prime },y+y^{\prime \prime }\right)$$

Here, ***w*** and ***h*** are the width and height of the image ***I***(***x***,***y***), respectively. For normalised correlation coefficient (CCoeff_Normed):(8)$$R\left(x,y\right)=\frac{\sum_{x^{\prime },y^{\prime }}{\left(T^{\prime }\left(x^{\prime },y^{\prime }\right)*I^{\prime }\left(x+x^{\prime },y+y^{\prime }\right)\right)}^{2}}{\sqrt{\sum_{x^{\prime },y^{\prime }}{\left(T^{\prime }\left(x^{\prime },y^{\prime }\right)\right)}^{2}*\sum_{x^{\prime },y^{\prime }}{\left(I^{\prime }\left(x+x^{\prime },y+y^{\prime }\right)\right)}^{2}}}$$

For the CCorr, CCorr_Normed, CCoeff, and CCoeff_Normed, the location (***x,y***) in ***R***(***x***,***y***) that has the maximum value is the position of the best match.

Another machine learning algorithm has been developed is based on principal component analysis (PCA), which calculates the principal components of the original data in order to maximise the variance [26,27]. The first principal component is equivalently the direction that maximises the variance of the projected data. The following principal component can be taken as a direction orthogonal to the previous principal component. PCA is commonly used for making predictive models [28,29,30], for dimensionality reduction [31,32] and classifications [33,34].

If we use ***pc*1** and ***pc*2** to represent the principal components of two images, then we can use the following equation to calculate the Euclidean distance between the two images:(9)$$d=\frac{1}{N}\sum_{i}\sqrt{\sum_{j}{\left(pc1\left(i,j\right)-pc2\left(i,j\right)\right)}^{2}}$$
where ***i*** is the ***i***th principal component, ***j*** is the ***j***th element in a principal component, and ***N*** is the total number of principal components. For PCA, the smaller the Euclidean distance, the more similar the two images, and the larger the Euclidean distance, the more different the two images. By calculating the Euclidean distance, we can understand how similar or different the two images are.

### 2.3. Measurement Procedure

Three sunscreens of a well-known brand with SPFs (Sun Protection Factors) of 20, 30, and 50+ were used in the study. Four skin sites on the volar forearm of healthy volunteers were chosen: three skin sites as test sites for sunscreens of SPF 20, 30, and 50+; the fourth skin site was used as a control, as illustrated in Figure 2. TEWL (trans-dermal water loss) and skin water content measurements were performed both before and after application of sunscreen. TEWL was measured using both AquaFlux and VapoMeter instruments, and skin water content was measured using the Epsilon permittivity imaging system.

All the measurements were performed on a healthy volunteer (male, 45–55, Asian), under normal ambient laboratory conditions of 20–21 °C and 40–50% RH. The volunteer was instructed to avoid excess water intake, and the measurements were performed in the morning. The volar forearm skin sites used were initially wiped clean with ETOH/H_2_O (95/5) solution. The volunteer was then acclimatised in the laboratory for 20 min prior to the experiments.

## 3. Results

### 3.1. Skin Water Loss Results

Figure 3 shows skin TEWL values of four volar forearm skin sites, before, 1 h after, and 2 h after the application of three different sunscreens. Each TEWL measurements were repeated five times. Figure 3A shows the average AquaFlux TEWL values and the corresponding standard deviations as error bars, which were found to decrease consistently on all three sunscreen skin sites after the application of the sunscreens, while the control site remained more or less the same. Figure 3B shows the VapoMeter TEWL values and the corresponding standard deviations as error bars, which were found to be not consistent. This was likely due large instrumental coefficient of variation (CV) of the VapoMeter, as shown in Figure 3C. CV was calculated as the ratio of standard deviation over the mean measured TEWL value as a percentage.

Figure 4 shows the corresponding TEWL value changes before, 1 h after, and 2 h after the sunscreen application, measured with the AquaFlux and VapoMeter. The AquaFlux results show that the TEWL values decreased 1 h after the sunscreen application, and continued to decrease 2 h after. This was the same for sunscreens with SPF 20, SPF 30, and SPF 50+. Of interest is the proportional decrease in TEWL with SPF number, with SPF 20 having the smallest TEWL decrease and SPF 50+ the largest. The control site had a small change with time. The VapoMeter results, however, did not show any consistent trend.

### 3.2. Skin Water Content Results

Skin water content measurements were also conducted along with the TEWL measurements on the same four volar forearm skin sites, before, 1 h after, and 2 h after the application of three different sunscreens. Skin water content measurements were repeated five times. Figure 5 below shows some example Epsilon permittivity images before and after applying sunscreen. Before application, all four skin sites were relatively dark, indicating low water content. After one hour, the three skin sites with the lotion applied became much brighter, indicating moisture content in the sunscreen. The sunscreen with SPF 20 had the highest brightness, and hence the possible highest moisture content. Conversely, the sunscreen with SPF 50+ had the lowest brightness, and hence the possible lowest moisture content. The control site remained more or less the same throughout.

In order to improve the Epsilon measurement accuracy, a template matching based on a selected region of interest (ROI) was used to analyse the results. Figure 6 shows the template matching results by using CCorr_Normed method, where Figure 6A shows the ROI in the first image, and Figure 6B,C show the best matching locations in the second image and the third image. The results show that the template matching can accurately relocate the ROI in the consequent images despite the similarity of the skin images.

Figure 7a,b shows the histograms of the analysed results corresponding to the images in Figure 5 by using the template searching method, again indicating an inverse proportional relationship, between skin hydration and SPF number. Figure 7c shows the ratio of skin hydration values over TEWL values, e.g., the ratio of and the Epsilon values shown in Figure 7a over the AquaFlux values shown in Figure 3a, which showed a consistent trend for the different skin sites.

### 3.3. Machine Learning PCA Results

Figure 8 shows the machine learning PCA results, where we calculated the Euclidean distances of skin control site after 2 h to different skin sites before and after the sun tan lotion application. The results show that the shortest Euclidean distance to skin control site after 2 h was itself then followed by the skin control site after 1 h, and skin site before the sun tan lotion application. The results also show consistent Euclidean distances to skin sites with SPF 50+, SPF 30, and SPF 20. In other words, it is possible to use PCA Euclidean distances to differentiate the effect of different sun tan lotions.

## 4. Discussion

The TEWL measurement results show the importance of the repeatability and accuracy of the instruments, especially when looking for small and subtle skin changes. Although the direct TEWL values did not directly reflect the small skin changes due to different sun tan lotion SPF, the relative changes before and after the application did reflect the corresponding changes. The skin capacitive image results show that it is possible to visualise the effects of different sun tan lotions. The ratio of skin water content values over skin TEWL values showed a consistent result, and therefore the combination of skin capacitive measurements and skin TEWL measurements can be a powerful tool for skin characterisations. The template matching algorithm works well on skin capacitive images and can improve the accuracy of the data analysis. The PCA is a powerful algorithm that can extract hidden features from the skin capacitive images. The PCA Euclidean distances can effectively differentiate the differences of different sun tan lotions. The next step in the future is to develop a PCA-based algorithm for skin capacitive image classifications and image searching. 

## 5. Conclusions

We present our latest study on the effect of sun tan lotion on skin by using skin TEWL measurements and skin water content measurements. We also developed machine learning algorithms for processing skin capacitive images. The overall results show that skin TEWL measurements and skin water content measurements are capable of detecting and quantifying subtle changes in skin condition after the application of different sun tan lotions. The TEWL results showed a decrease in TEWL after the sunscreen application compared to the control site, with SPF 50+ having the most effect and SPF 20 the least. The TEWL results also show the importance of instrument repeatability and accuracy when measuring small changes. The skin water content measurements by using skin capacitive imaging showed a significant increase in skin water content after sunscreen application, with SPF 20 increasing the most and SPF 50+ the least. The ratio of skin water content values over skin TEWL values could be a reliable good indicator for skin status.

For future work, we will conduct a larger scale study with more volunteers in order to observe the effects of age, gender, skin colour, and so on. We will also apply this approach to other skin products, such as soaps, shampoos, washing liquids, skin creams, and lotions in order to differentiate the small differences of different products. We will also continue to develop machine learning algorithms for analysing skin capacitive images, such as image classifications and image searching.

## Figures and Tables

**Figure 1 sensors-22-03595-f001:**
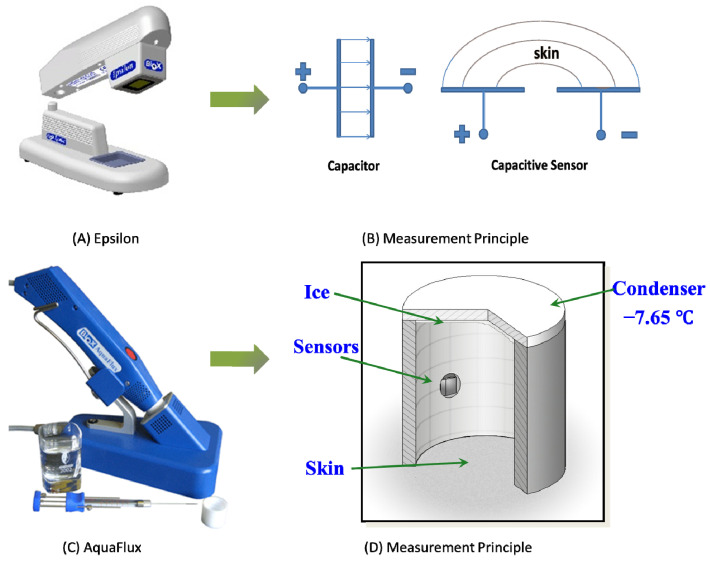
Epsilon permittivity imaging system (**A**) and its measurement principle (**B**); AquaFlux (**C**) and its measurement principle (**D**).

**Figure 2 sensors-22-03595-f002:**
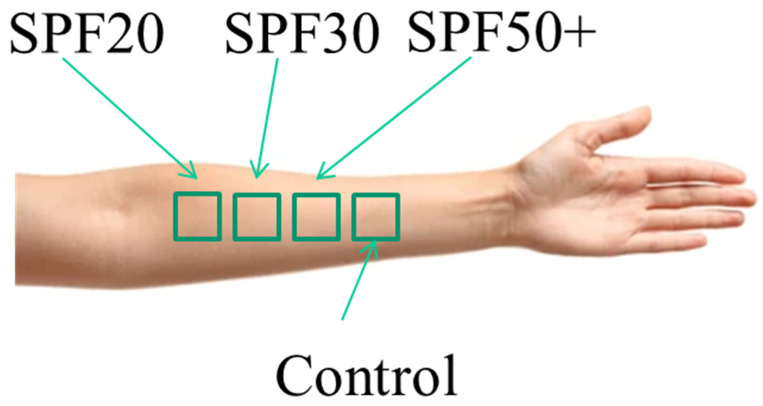
Four skin sites on the volar forearm. Three were test sites for sunscreen with SPF 20, 30, and 50+. The fourth skin site was chosen as the control.

**Figure 3 sensors-22-03595-f003:**
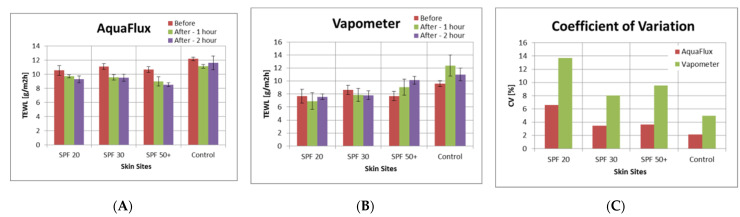
The TEWL values of four volar forearm skin sites before and after the sun tan lotion application, measured by AquaFlux (**A**) and VapoMeter (**B**), and their coefficient of variation (**C**).

**Figure 4 sensors-22-03595-f004:**
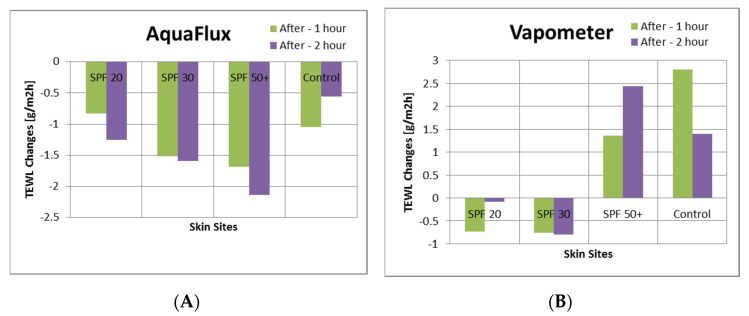
The TEWL value changes before and after sunscreen application, measured by AquaFlux (**A**) and VapoMeter (**B**).

**Figure 5 sensors-22-03595-f005:**
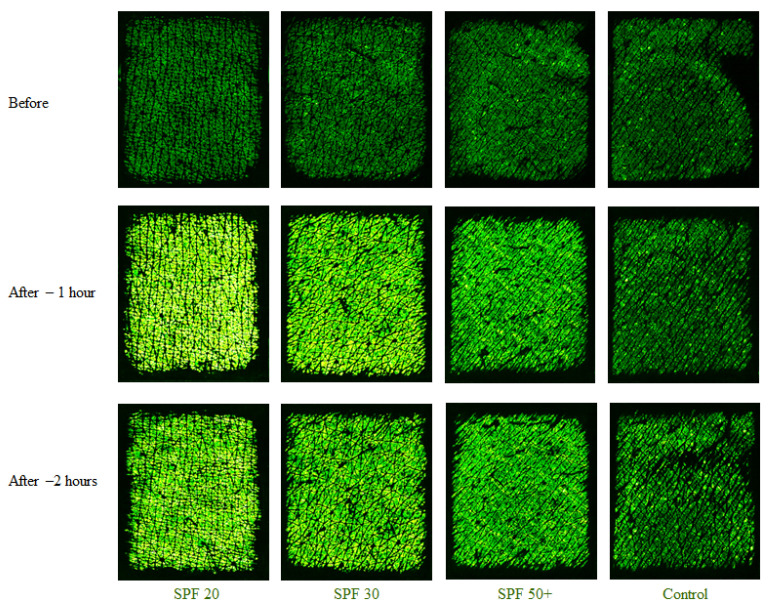
Epsilon permittivity images of four volar forearm test sites before, 1 h after, and 2 h after application of sunscreen. SPF 20, SPF 30, and SPF 50+ sunscreen was applied on three skin sites. The fourth skin site was used as a control.

**Figure 6 sensors-22-03595-f006:**
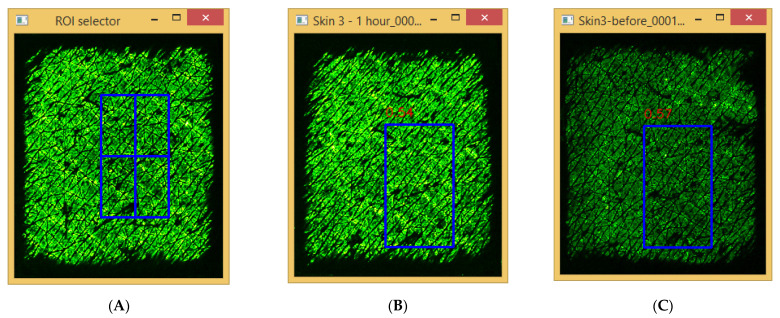
The region of interest (ROI) shown as crossed rectangle in the first image (**A**), and the best matching locations in the second image (**B**) and the third image (**C**), shown as empty rectangles.

**Figure 7 sensors-22-03595-f007:**
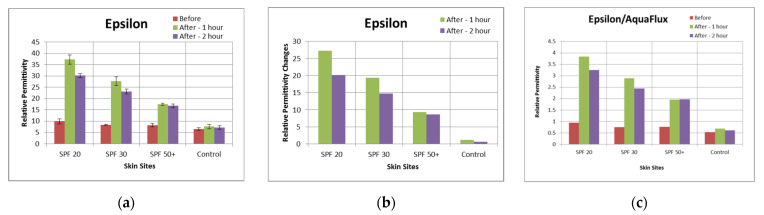
Epsilon mean permittivity readouts of the four skin sites before and after sunscreen application (**a**), the corresponding change in permittivity at each site (**b**), and the ratio of AquaFlux values over Epsilon values (**c**).

**Figure 8 sensors-22-03595-f008:**
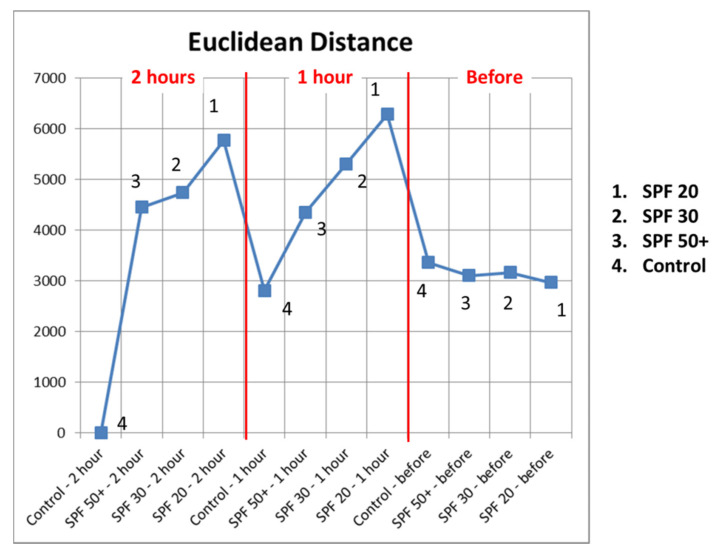
The Euclidean distances of skin control site after 2 h to different skin sites before and after the sun tan lotion application, where skin site 1: SPF 20, 2: SPF 30, 3: SPF 50+, and 4: control.

## Data Availability

All the data generated during the study are available upon request.

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
