# Peer review of "The Effect of Sun Tan Lotion on Skin by Using Skin TEWL and Skin Water Content Measurements"

_sensors, 2022, doi:10.3390/s22093595_

Round 1

Reviewer 1 Report

Evaluating the SC characterization is important to the cosmetic properties with simple method for example with sensor. This manuscript presented some test results with two types of skin measurement device. As a research paper, some comments are as follows.
1/ the purpose of this study is not very clear, though the author mentioned for developing a new SC barrier characterization method by using both TEWL and capacitive imaging. However it is not clear introduced that those two method’s disadvantages in using, for example, accuracy, repeatability or stability? This work may go any steps further when combine the two methods. 
2/in section 2, the author introduced many searching methods in order to get the best matching results, do these methods apply to the test results? What benefits be got and meet the goal? Why select two brands of TEWL meter? They have the different test results for the same site measurement that also indicate the calibration problem existed or the related standard of TEWL. 
3/in section 3, the measurement results of this work should compare with other’s work especially using the two methods.  Comparing the two commercial TEWL devices with different SPF, they do not show any consistent, why? In the Epsilon test, the results may effect by the poor skin contact with the sensor, it is unavoidable. So that, how many tests have been done in fig 6? Does the error be analyzed?

Author Response

We have revised the manuscripte according to reviewers comments, revised text are highlighted in red colour. The following are the details.

1/ the purpose of this study is not very clear, though the author mentioned for developing a new SC barrier characterization method by using both TEWL and capacitive imaging. However it is not clear introduced that those two method’s disadvantages in using, for example, accuracy, repeatability or stability? 

Refined the purpose of the study, and added text and references about the accuracy and repeatability.

2/in section 2, the author introduced many searching methods in order to get the best matching results, do these methods apply to the test results? What benefits be got and meet the goal? Why select two brands of TEWL meter? They have the different test results for the same site measurement that also indicate the calibration problem existed or the related standard of TEWL. 

Recalculated Epsilon results using template searching. Also added error bars to the TEWL and Epsilon results. Updated Figure 3, 6, 7. 

3/in section 3, the measurement results of this work should compare with other’s work especially using the two methods.  Comparing the two commercial TEWL devices with different SPF, they do not show any consistent, why? In the Epsilon test, the results may effect by the poor skin contact with the sensor, it is unavoidable. So that, how many tests have been done in fig 6? Does the error be analyzed?

Two commercial TEWL devices do not show any consistent because of their measurement accuracy and repeatability. Errors bars have been added to the TEWL results.  Epsilon contact has been explained by adapting spring loaded mechanism. Fig 6 has been tested 6 times, and error bars were added.

Reviewer 2 Report

The proposal has good approaches to the latest study on the effect of suntan lotion on the skin by using skin TEWL and skin water content measurements. Authors have also developed Machine Learning algorithms for processing the skin capacitive images.

Make a brief introduction after "2. Materials and Methods" which is found in this section. Same in "3. Results"

Vectorize the images to see the details of all pictures.

Check the English by a native speaker.

Line 106 can be justified also with the following references regarding PCA analysis: A Study of Movement Classification of the Lower Limb Based on up to 4-EMG Channels; Support Vector Machine-Based EMG Signal Classification Techniques: A Review; Public space accessibility and machine learning tools for street vending spatial
categorization

I have two major concerns:

1.- The innovation of the article, but clearly in the conclusion and abstract sections the findings, if those are quantitative it will be better.

2.-Perform a table by comparing this research vs already reported in order to highlight the findings.

Author Response

We have revised the manuscripte according to reviewer's comment, all the changes are highlighted in red colour, the following are the details.

Make a brief introduction after "2. Materials and Methods" which is found in this section. Same in "3. Results"

Done

Vectorize the images to see the details of all pictures.

Figure quality improved. 

Check the English by a native speaker.

Done

Line 106 can be justified also with the following references regarding PCA analysis: A Study of Movement Classification of the Lower Limb Based on up to 4-EMG Channels; Support Vector Machine-Based EMG Signal Classification Techniques: A Review; Public space accessibility and machine learning tools for street vending spatial
categorization

Reference added.

I have two major concerns:

1.- The innovation of the article, but clearly in the conclusion and abstract sections the findings, if those are quantitative it will be better.

Revised.

2.-Perform a table by comparing this research vs already reported in order to highlight the findings.

No previous studies on effects of sun tan lotion different SPFs on skin TEWL measurements and skin hydration measurements.

Reviewer 3 Report

The manuscript presents a study on the effect of sun tan lotion on skin by using skin trans-epidermal water loss (TEWL) measurements and skin water content measurementsl, being the aim to develop a new stratum corneum barrier characterisation method by using both TEWL and capacitive imaging measurements. 
The results show that the TEWL values decreased after the sun tan lotion application.
I find the topic interesting and being worth of investigation and the document is well strucutred, organized, fluidly written, has enough background information, the methodology followed is clearly explained, formulas are correct, the results are clearly presented.
Although I propose the following comments/suggestions:
- The abstract is poorly descriptive of the content, it should be better organized: problem, motivation, aim, methodology, main results, further impact of those results.
- keywords should be in alphabetical order.
- I strongly suggest authors from refraining using personal pronouns such as "we" and "our" throughout the text and I encourage them to write it in an impersonal form of writing.
- There is no description of the sample used characterization and analysis of possible influencing factors (e.g. age, sex, skin color, ...)
- The charts at the results only show mean values, no standard deviation or error are presented, Bland Altmann degree of similarity should be used instead
- The study limitations are not present at discussion section and neither relationship with previous research is made.
- No further research is proposed at conclusions.

Author Response

We have revised the manuscript according to reviewer's comments, changes are highlighted in red colour. The following are the details.

- The abstract is poorly descriptive of the content, it should be better organized: problem, motivation, aim, methodology, main results, further impact of those results.

Abstract revised.
- keywords should be in alphabetical order.

Done
- I strongly suggest authors from refraining using personal pronouns such as "we" and "our" throughout the text and I encourage them to write it in an impersonal form of writing.

We disagree with this, as "we" and "our" are commonly used in scientific journal papers.

- There is no description of the sample used characterization and analysis of possible influencing factors (e.g. age, sex, skin color, ...)

Information added.
- The charts at the results only show mean values, no standard deviation or error are presented, Bland Altmann degree of similarity should be used instead

Standard deviation is added to the chart.
- The study limitations are not present at discussion section and neither relationship with previous research is made.

No previous studies on the effect of sun tan lotion SPF on TEWL and water content measurement, but more discussions are added.
- No further research is proposed at conclusions.

Future work added in conclusions.

Round 2

Reviewer 1 Report

The authors have revised the manuscript partly. It is in fact that design of the experiment was not very appropriate and rigorous, for example, adapted two TEWL device, and volunteers' drinking water control, test time selection in the method, etc.

Author Response

We have added extra information to address the following reviewer's comments, all changes are tracked.

The authors have revised the manuscript partly. It is in fact that design of the experiment was not very appropriate and rigorous, for example, adapted two TEWL device, and volunteers' drinking water control, test time selection in the method, etc. (Extra information added)

Reviewer 2 Report

The manuscript has been greatly improved. It can be accepted for publication.

Author Response

Thank you very much for the positive comments!

Reviewer 3 Report

The manuscript presents a study on the effect of sun tan lotion on skin by using skin SC trans-epidermal water loss (TEWL) and skin water content measurements.

Despite the authors have improved significantly the manuscript acommodating the reviewers comments and suggestions, I still have the following points to be addressed:

  • I disagree with the authors in the usage of personal pronouns such as "we" and "our" throughout the text, which is a inelegant form of scientific writing.
  • The authors do not show statistics at the results on the impact of  possible influencing factors (e.g. age, sex, skin color, ...)

Author Response

We have revised the manuscript, and all changes are tracked. The following are details:

Despite the authors have improved significantly the manuscript acommodating the reviewers comments and suggestions, I still have the following points to be addressed:

  • I disagree with the authors in the usage of personal pronouns such as "we" and "our" throughout the text, which is a inelegant form of scientific writing.

We still think it is fine, take the following latest Sensors publication, it uses "we" and "our".

https://www.mdpi.com/1424-8220/22/1/352

  • The authors do not show statistics at the results on the impact of  possible influencing factors (e.g. age, sex, skin color, ...)

This paper is the first step, we have modified the future work, to make it next step to conduct the study for larger volunteers in order to observe the effects of age, gender, skin colour and so on.